# Determination of the Influence of Multiple Closed Recycling Loops on the Property Profile of Different Polyolefins

**DOI:** 10.3390/polym14122429

**Published:** 2022-06-15

**Authors:** Johanna Langwieser, Andrea Schweighuber, Alexander Felgel-Farnholz, Christian Marschik, Wolfgang Buchberger, Joerg Fischer

**Affiliations:** 1Competence Center CHASE GmbH, Altenberger Straße 69, 4040 Linz, Austria; christian.marschik@chasecenter.at; 2Institute of Polymeric Materials and Testing, Johannes Kepler University Linz, Altenberger Straße 69, 4040 Linz, Austria; joerg.fischer@jku.at; 3Institute of Analytical and General Chemistry, Johannes Kepler University Linz, Altenberger Straße 69, 4040 Linz, Austria; andrea.schweighuber@jku.at (A.S.); alexander.felgel-farnholz@jku.at (A.F.-F.); wolfgang.buchberger@jku.at (W.B.)

**Keywords:** mechanical recycling, closed-loop, polyolefins, circular testing, polymer degradation

## Abstract

In a circular economy, polymeric materials are used in multiple loops to manufacture products. Therefore, closed-loops are also envisaged for the mechanical recycling of plastics, in which plastic is used for products that are in turn collected and reprocessed again and again to make further products. However, this reprocessing involves degradation processes within the plastics, which become apparent through changes in the property profile of the material. In the present paper, the influence of multiple recycling loops on the material properties of four different polyolefins was analyzed. Two different closed-loop cycles with industrially sized processing machines were defined, and each polyolefin was processed and reprocessed within the predefined cycles. For the investigation of the effect of the respective loops, samples were taken after each loop. The samples were characterized by high-pressure liquid chromatography coupled to a quadru-pole time-of-flight MS, high-temperature gel permeation chromatography, melt flow rate measurements, infrared spectroscopy, differential thermal analysis, and tensile tests. With increasing number of processing loops, the tested polyolefins showed continuous material degradation, which resulted in significant changes in the property profiles.

## 1. Introduction

Nowadays, 60% of plastic products have a use phase between one and 50 years, which is the lapse of time a product is in use until it becomes waste [1]. The life cycle of each product is different, which means that the consumption and production quantity does not match. In 2018, 29.1 million tons post-consumer plastic waste were collected in Europe. Of those, around 42% were used for energy recovery, 33% were recycled and 25% were brought to landfills [1,2]. In 2018, the highest recycling rate of plastic packaging waste was recorded in Lithuania (69.3%) followed by Slovenia (60.4%) and Bulgaria (59.2%) [3].

On the European market over 600 companies work in the plastics recycling industry with only five countries covering 67% of the total recycling capacity (France, Germany, Italy, Spain, UK). 80% of the total recycling capacity in Europe is covered by polyethylene terephthalate (PET) and polyolefins (polyethylene and polypropylene) [4]. In 2018, Europe had a processing capacity to recycle 1.7 million tons of rigid polyethylene high-density (PE-HD) and polypropylene (PP) (1.2 million tons for post-consumer rigid polyolefins, 0.5 million tons for pre-consumer material). The 1.2 million tons of post-consumer rigid polyolefins can yield 0.8 million tons of post-consumer recyclates (PCR) with a reject of around 25% [5].

Even though a lot of material is already recycled, around 75% of the used plastics in Europe are either thermally recovered or landfilled [1]. To aim for higher recycling rates, different approaches are defined all over the world. The 2030 Agenda for Sustainable Development, for example, was accepted by member states of the United Nations in 2015. 17 Sustainable Development Goals (SDGs) were settled on, in which the 12th Goal is “To ensure sustainable consumption and production patterns”, one target is to reduce waste generation by prevention, reduction, recycling and reuse [6].

For a circular economy for plastics three common methods are used: (i) thermal, (ii) chemical and (iii) mechanical recycling. Thermal recycling (i) is the incineration of waste with energy recovery. The energy in the form of heat is used to produce steam and in turn to generate electricity. Chemical recycling (ii) is the process of converting the polymer chains to low molecular weight compounds or, if possible, into the original plastic monomer [7,8]. This paper focuses on the third option.

Mechanical recycling (iii) uses physical processes such as milling, washing, sorting, drying, re-granulating, and compounding to transform plastic waste into useable polymer feedstock. It is a well-established and an economical way to reprocess different materials. Mechanical recycling keeps the polymer molecule mostly intact. The material source can be divided into pre-consumer and post-consumer, whereas pre-consumer feedstock is usually clean and of a single type or at least of a known composition. Post-consumer waste is often strongly contaminated and needs additional treatment steps [9,10,11].

An objective to be pursued by mechanical recycling is closed-loop recycling where material is used for products which in turn will be collected and reprocessed for similar products repeatedly. Within this closed-loop the material is processed multiple times and each step has a significant influence on the material properties. Polyolefins degrade in a physical or chemical way triggered by light, heat, humidity, chemicals or bioactive substances. This breakdown becomes noticeable by a change in the material’s mechanical, optical or electrical properties (e.g., crazes, cracks, erosion, discoloring). Within the material, degradation happens by chain scission, crosslinking, chemical transformations or by formation of new functional groups [12,13].

Numerous studies focused on the influence of multiple recycling cycles on the quality of polyolefins [14,15,16,17,18,19,20,21,22,23]. These analyses can be classified into two groups. On the one hand, studies investigated closed-loops using only one processing technique such as Pinheiro et al. [14] and Canevarolo [15] who looked at the mechanical and thermo-oxidative degradation of PE-HD and PP using a 30 mm twin-screw extruder with different processing conditions and screw profiles. The material was reprocessed five times and characterized by their residence time distribution which was measured in-line at the die exit but only during the first extrusion, by IR-spectra using thick hot-pressed films, by molecular weight distribution using gel permeation chromatography. The results coming from the analyses displayed degradation in the form of crosslinking for PE-HD and chain scission for PP. Bernardo et al. [16] investigated the properties of different PE-HD and PE-LD grades after ten injection molding cycles where chain scission and crosslinking was observed. In their study the reprocessed material was mixed with virgin material in every loop. With multiple processing the injection molded specimens changed color which reflects the formation of conjugated double bonds. Other works looked at specific applications for closed-loops such as films for greenhouses [17], discontinuous carbon fiber polypropylene composites [18] or in general polyethylene-like materials [19].

On the other hand, studies looked at loops with multiple degradation, but not multiple processing steps, such as Jansson et al. [20] who subjected two post-consumer PPs to repeated extrusion (using a 19 mm diameter extruder) and thermo-oxidative ageing of the extruded sheets separately. The materials showed a decrease of their oxidative induction temperature with every aging and extrusion step and showed changes in their strain-at-break behavior. Furthermore, other publications used smaller lab sized equipment such as Oblak et al. (L/D ratio of 16/40), Jin et al. [22] (L/D ratio of 16/40) and Schweighuber et al. [23] (L/D ratio of 5/14).

The objective of this study was the determination of the influence of multiple recycling cycles on the material properties of selected polyolefins, using industrially sized processing machines. Four virgin materials including a high-density polyethylene, a low-density polyethylene, and two polypropylenes with different melt flow rates were processed in two closed-loop cycles. The first cycle involved an extrusion and granulation step based on an underwater pelletizing system. The second cycle was designed to simulate an additional product phase, which was realized by injection molded multipurpose specimens. The material was extruded and granulated, injection molded, and lastly milled. Both closed-loop cycles were repeated six times. Properties of the samples were characterized by high-pressure liquid chromatography coupled to a quadrupole time-of-flight MS (HPLC-QTOF-MS), high-temperature gel permeation chromatography (HT-GPC), melt flow rate (MFR) measurements, infrared (IR) spectroscopy, differential thermal analysis, and tensile tests.

## 2. Materials and Methods

### 2.1. Materials

Four different materials were used: (i) a polyethylene high-density (PE-HD) with a melt flow rate (MFR) of 4 g/10 min (190 °C/2.16 kg), which is a typical injection and compression molding grade. The material is typically used in industrial food and transport packaging; (ii) a polyethylene low-density (PE-LD) with an MFR of 1.9 g/10 min (190 °C/2.16 kg), which is industrially applied in film extrusion for different packaging; (iii) a polypropylene injection molding type, this material has an MFR of 12 g/10 min (230 °C/2.16 kg) and is used for house ware and thin wall packaging; (iv) a polypropylene with an MFR of 4 g/10 min used for thermoforming packaging. An overview of the materials including their abbreviations used in this work is depicted in Table 1, the structural formulas of the materials can be seen in Figure 1.

### 2.2. Closed-Loop Recycling Cycles

To analyze the effect of multiple processing steps on the properties of our materials, two closed-loop recycling cycles were simulated. The first cycle (further referred to as CL-E) used a 35 mm single-screw extruder (SML Maschinengesellschaft mbH, Lenzing, Austria) with a maximum mechanical power of 38.4 kW provided by the electrical drive unit and a maximum screw speed of 434 rpm. At the extruder head an EUP 50 under-water pelletizing system (Econ GmbH, Weisskirchen, Austria) was installed to granulate the extruded melt. To control the temperature of the extruder, it was equipped with six heating zones, as illustrated in Table 2, the feed opening (FO) and the heating zones one to three (Z1–Z3) are located directly on the barrel of the extruder, whereas adapter one (AD1) and adapter two (AD2) are responsible for heating the connections between the under-water pelletizing system and the extruder. The feed housing was water-cooled whereas the remaining barrel sections were cooled by forced air to avoid excessive heat generation. The materials were processed with a screw speed of 100 rpm which resulted in an output of approximately 25 kg/h. The melt temperatures for the PE- and PP-types were 200 °C and 230 °C, respectively. The bypass and the die plate were set at a temperature of 180 °C, the cooling water temperature was held beneath 30 °C. The granulation speed was around 1500 rpm, the dryer speed was around 800 rpm. The material was extruded and granulated six times and samples were taken before each extrusion step (marked with a red box in Figure 2).

The second cycle combined multiple processing steps by adding an injection molding step after granulation (further referred to as CL-EI). The main purpose of this modification was to additionally simulate the product processing phase of the material. In addition to the extrusion and the granulating step with the same processing parameters, the material within this cycle was injection molded to multipurpose specimens (ISO 17855-2 [24]) using an injection molding machine victory 60 (Engel Austria GmbH, Schwertberg, Austria) and milled with a granulator S-Max 2 Plus (Wittmann Battenfeld GmbH, Kottingbrunn, Austria) at 27 rpm. The temperature program for the injection molding machine is shown in Table 3 whereas Z1 to Z4 are abbreviations for the heating zones on the plasticizing unit. Additional processing parameters for the injection molding machine can be seen in Table 4. In total the CL-EI was repeated six times, sample collection is marked with a red box in Figure 3.

### 2.3. Characterization Methods

For both recycling cycles, selected properties were measured for all materials after each loop, including the consumption of stabilizers, the weight average molar masses and molar mass distributions, the melt flow rate, the carbonyl index, the degree of crystallinity and stress-strain diagrams.

#### 2.3.1. High-Pressure Liquid Chromatography (HPLC)

To identify the consumption of the non-oxidized contents of the processing stabilizer, the secondary antioxidant Irgafos 168, and of the radical scavengers, the primary phenolic antioxidant Irganox 1010, an Agilent 1260 high-pressure liquid chromatograph coupled to an Agilent 6510 quadrupole time-of-flight MS (HPLC-QTOF-MS) with an electrospray ionization (ESI) source (Agilent Technologies Inc., Santa Clara, CA, USA) was used. For this purpose, the HPLC was equipped with a Kinetex C18 separation column (Phenomenex Inc., Aschaffenburg, Germany) and a C18 guard column (Phenomenex Inc., Aschaffenburg, Germany). Furthermore, tributyl phosphate was added to avoid stabilizer loss during sample preparation, and Cyanox 1790 was used as an internal standard.

#### 2.3.2. Gel Permeation Chromatography (GPC)

A high-temperature gel permeation chromatograph (HT-GPC, PolymerChar S.A., Valencia, Spain) equipped with an IR 5 detector was used to obtain molar mass distributions (MMD) and weight average molar masses (M_w_) of all collected samples from both loops. To dissolve the samples trichlorobenzene stabilized with butylated hydroxytoluene was used, and heptane was added as a flow marker.

#### 2.3.3. Melt Flow Rate (MFR)

MFR measurements are single point measurements at low shear rates (depending on the measurement weight) which are often used for quality control of incoming materials. Since it is an easy and quick tool, it was used to compare each material sample of each cycle. For this purpose, the Aflow (ZwickRoell GmbH and Co. KG, Ulm, Germany) was used. The measurements were performed according to ISO 1133 [25] with a measurement weight of 2.16 kg and measuring temperatures of 190 °C for PE and 230 °C for PP. For each run about three grams of material were filled into the preheated cylinder and compacted two to three times. After five minutes of warm up time the measurement started automatically, and the piston pressed the molten material through the die. Five samples were cut from the middle strand and weighed. For each sample at least two measurements were carried out.

#### 2.3.4. Infrared Spectroscopy (IR Spectroscopy)

The IR spectra were recorded using the Fourier transform infrared (FTIR) spectrometer Spectrum Two (Perkin Elmer Inc., Waltham, MA, USA). The measurements were conducted within the wavenumbers of 500 cm^−1^ to 4000 cm^−1^. Each sample was measured at least twice. To evaluate the degradation within the materials, the carbonyl index (CI) was calculated by comparing the absorption peak for a carbonyl group with an existing reference peak of the investigated polymer. The carbonyl peak can be detected in the wavenumber range from 1850 to 1650 cm^−1^. For the CI evaluation of PE, the CH_2_ scissoring vibrations at approximately 1463 cm^−1^ and for PP, the CH_2_ and CH_3_ vibrations at around 1459 cm^−1^ were chosen. To calculate the CI, the area underneath the carbonyl peak was divided by the area underneath the reference peak as depicted in Equation (1) [26].
(1)$$CI=\frac{A_{1850-1650\;cm^{-1}}}{A_{reference}}\;,$$

#### 2.3.5. Differential Thermal Analysis (DTA)

The heat of fusion $\Delta H_{melt}$ and the specific heat capacity were measured using a DSC 4000 (Perkin Elmer Inc., Waltham, MA, USA). Heating and cooling rate was 10 K/min from 30 °C to 230 °C. The mass of the specimens was 5.0 ± 0.5 mg. All measurements were performed in an air atmosphere. The degree of crystallinity was calculated based on Equation (2):(2)$$\alpha =\frac{\Delta H_{melt}}{\Delta H_{100\%}}\times 100\%\;,$$
where $\Delta H_{100\%}$ refers to the heat of fusion of 100% crystalline polymer. For polyethylene, it is 293 J/g and for polypropylene it is 207 J/g [27]. For the PE-I, $\Delta H_{melt}$ was evaluated between 70 and 150 °C, for PE-E, in the range of 60 and 125 °C, and for both PP grades between 120 and 180 °C.

#### 2.3.6. Tensile Test

The stress-strain diagrams were recorded using the universal testing machine Z020 (ZwickRoell GmbH and Co. KG, Ulm, Germany). Multipurpose specimens were tested according to ISO 527-2 [28]. At the beginning of the measurement an initial load of 0.1 MPa was reached. All specimens were measured with a testing speed of 1 mm/min in the strain region of 0.05% to 0.25% and then tested with a testing speed of 50 mm/min until failure via break occurred. For each sample at least ten measurements were conducted.

## 3. Results

This chapter discusses the results of our experimental study. To differentiate the two loops, grey lines for the extrusion-cycle (CL-E) and red lines for the extrusion-injection molding cycle (CL-EI) were used. In this paper the single point data is presented, the mentioned multi point data can be found in the Appendix A.

### 3.1. Consumption of Stabilizers

According to the data sheets [29,30], Irgafos 168 (IF168) is a hydrolytically stable secondary antioxidant, which reacts during processing with hydroperoxides formed by autoxidation and thereby prevents polymer degradation induced by processing, protects the primary antioxidants, and thus extends the performance of primary antioxidants. Irganox 1010 (IX1010) is a sterically hindered primary phenolic antioxidant which protects organic substrates against thermo-oxidative degradation. Both stabilizers can be used in polyolefins.

Figure 4a depicts the amount of non-oxidized stabilizers left inside PE-I over the reprocessing loops. Compared to PE-I with about 0.27 w% of IF168 and about 0.02 w% of IX1010 within the virgin samples, no such stabilizers (IF168 and IX1010) were found in PE-E. For PE-I, only for the CL-EI a substantial and continuous reduction by increased processing of IF168 was obtained. IF168 was depleted after the fifth loop. While for the CL-E the IX1010 values stayed constant, for the CL-EI, an initial drop after the first loop and constant values afterwards were detected.

In Figure 4b,c, the stabilizer content in both PP grades after each loop is illustrated. PP-E possessed with approximately 0.35 w% for IF168 and 0.16 w% for IX1010 in the initial virgin state roughly double the stabilizer content than PP-I. The stabilizer contents of both materials showed a significant drop in both cycles. The remaining intact amounts of IF168 and IX1010 were in general lower after CL-EI than after CL-E. For both PPs, IF168 was sooner depleted than the IX1010, depicted in full IF168 loss for PP-E in the CL-EI after the fourth loop, for PP-I in the CL-E after the third loop, and for PP-I in the CL-EI after the first loop.

### 3.2. Molar Mass Distributions (MMD) and Weight Average Molar Mass (M_w_)

In Appendix A, the changes in molar mass distribution (MMD) curves and in Figure 5a the weight average molar mass (M_w_) of PE-E after each loop of both processing cycles, CL-E and CL-EI, are illustrated. Overall, for the CL-E no significant changes in the distribution were observed. For the CL-EI the shape of the distribution curve changed to a flatter and wider one with increased processing. Just as Jin et al. [22] showed, also in our investigations of the CL-EI, a shift in the peak of the MMD to smaller values and a shape change at higher molecular weights was observed. These changes in the distribution curves suggest that the molecular weight increased due to crosslinking.

To acquire a better picture of the overall changes measured via GPC, M_w_ was evaluated. It stayed rather constant after the CL-E with values around 91,200 g/mol but significantly increased in course of the CL-EI up to ca. 111,000 g/mol. Degradation in polyethylene can lead to crosslinks between the polymer chains which stems from the reaction of freshly cut chains due to shear and temperature with the terminal vinyl unsaturation [14]. This results in an increase in molecular mass by branching and crosslinking.

Appendix A depicts the MMD and Figure 5b M_w_ of PE-I. No significant changes in the shape of the MMD could be observed for either the CL-E or the CL-EI. The weight average molar mass dropped for both cycles during the first processing step and remained approximately constant between 90,000 and 95,000 g/mol afterwards.

In Appendix A and Figure 5c, with increasing loops a shape change of the MMD in both processing cycles of PP-E was illustrated. Just as mentioned above, also for PP-E, the curves shifted to lower values. However, in contrast to PE-E, increasing peaks were detected, which was also discovered by Canevarolo et al. [15]. This change led to the assumption, that due to the thermo-mechanical degradation the longer chains break, and the resulting shorter chains are added to the middle part of the MMD curve. With increasing processing, the amount of the shorter chains got bigger, and the curve shifted upwards.

A significant decrease in M_w_ starting with about 350,000 g/mol in the virgin material was observed whereas weight average molar masses of the CL-EI were lower with the last and lowest value at 213,000 g/mol comparted to 90,000 g/mol for the CL-E. This behavior over all is an indicator for chain scission in a material, which is the preferred degradation behavior of PP. The chains continuously broke due to thermal and mechanical influences and thereby decreased their molecular mass. This decrease led to a narrowing and a peak increase of the MMD curve.

Appendix A shows the MMD and Figure 5d the M_w_ of PP-I. Again, the typical degradation behavior of PP can be observed, and the molecular masses decreased continuously in both cycles, which led to an increase, a shift to lower values and a narrowing of the MMD. The M_w_ for the CL-EI was continuously lower than for the CL-E with the lowest value of 144,000 g/mol after six loops of CL-EI and of 210,000 g/mol after six loops of CL-E.

### 3.3. Melt Flow Rate (MFR)

The MFR is a single point measurement and a measure for the fluidity of a material at low shear rates [31]. The increase in molecular mass illustrated above means longer and more connected chains which results in hindered movement of the polymer and thus leads to a higher viscosity and lower MFR. The review by Yin et al. [10] states that based on the MFR, PP grades have good thermal stability and degrade later than PE-HD or PE-LD types, which show imminent MFR reductions and thus have poor thermal stability. In our investigations the opposite behavior was obtained.

The MFR results of PE-E and PE-I can be seen in Figure 6a,b. The CL-E hardly influenced the viscosity of the PE types while the CL-EI led to a decrease of MFR values for both, which may have been caused by aforementioned degradation effects. According to these measurements, the increased processing using the extrusion-injection molding cycle influenced the viscosity of the PE types more than just processing by extrusion, which was expected and was also detected by the methods discussed before.

Both PP grades showed a significant MFR increase after both loops (see Figure 6c,d). PP commonly shows degradation in the form of chain scission as mentioned above. Due to shorter chains, the molecular mass decreased (see Figure 5) and the molecules were entangled to a lesser extent. This elicits better movement in the fluid state and the chains glide apart. Thus, in the same time a higher amount of material can flow through the measurement die, which resulted in an increased MFR. PP-E illustrated only a slight increase up until the third loop, where the MFR values started to increase in a non-linear way. PP-I shows a decrease of viscosity starting from the first loop.

### 3.4. Infrared Spectra (IR Spectra)

Appendix A depicts the evaluated IR data of the PE-E used in this study. In Appendix A characteristic peaks of PE can be distinguished [32]:3000 and 2840 cm^−1^: symmetric and antisymmetric stretching vibrations of CH_2_ groups and1463 cm^−1^: scissoring vibrations of CH_2_ groups.

In Appendix A additional peaks at 3500–3000 cm^−1^ and 1700–1500 cm^−1^ were detected. These peaks may stem from the lubricant (Erucamide according to the data sheet) used. Due to the increased processing, the lubricant seemed to migrate out of the material onto its surface. Owing to the carbonyl group in the used lubricant, the CI was distorted which led to higher and more stable values until the lubricant wore off (see grey curve in Figure 7a).

In Appendix A the additional peaks caused by the lubricant disappear. This indicates that due to the excessive processing in CL-EI, the lubricant migrated completely to the surface and was removed. With further processing the carbonyl peaks increased, which suggested that the material oxidized with further processing, as shown in Figure 7a in red.

PE-I showed next to the characteristic peaks’ significant changes in the wavenumber range related to the absorption of the carbonyl groups. Since no further peaks can be observed in the spectrum, the increase of the carbonyl groups presumably stemmed from oxidation due to degradation. This behavior can be seen in Figure 7b. Processing according to the CL-E led to a continuous increase of the CI up to the sixth loop. The CL-EI influenced the material by a sudden increase after the second loop and a further continuous decrease.

The IR spectra of the PP grades can be seen in Appendix A. PP exhibited the following characteristic peaks [32]:3000 to 2840 cm^−1^: symmetric and antisymmetric stretching vibrations of CH_2_ and CH_3_ groups and1459 cm^−1^ and 1376 cm^−1^: bending vibrations of CH_2_ and CH_3_ groups.

For PP-E (Figure 7c), the CI reached its highest value for both cycles after the first loop and decreased furthermore. After the fifth loop of the CL-E, the CI of the material started to increase again.

For PP-I, distinct CI values for the CL-E could not be distinguished due to the appearance of additional peaks which may have been present due to a lubricant used, such as estimated for PE-E before. For the CL-EI a continuous increase of the CI could be seen in Figure 7d.

### 3.5. Melting Behavior and Degree of Crystallinity

Appendix A illustrate the melting peaks of both polyethylene grades with no significant trends. Additionally, the degrees of crystallinity are shown in Figure 8a,b. According to literature [13], typical crystallinity values for PE-LD range between 40 and 50%, whereas PE-HD lies between 60 and 80%, which is in agreement for the tested materials PE-E and PE-I, respectively. For both PE grades, the CL-EI values were lower than the values of the CL-E. Furthermore, the standard deviations of PE-I were bigger than the PE-E’s. Jin et al. [22] and Oblak. et al. [21] discovered rather constant degrees of crystallinities up until the 40th loop and 20th loop, respectively. This is in agreement with our measurements concerning PE, where no significant trend was obtained.

Appendix A show the melting peaks of both PP grades, again with no significant trend. Furthermore, the degrees of crystallinity are depicted. Typical values of the crystallinity range between 30 and 60% [13]. Hence, within our investigations some results exceeded the typical range. In Figure 8c the PP-E shows increase and decrease in a random manner for both cycles even though the CL-E depicts stronger fluctuations than the CL-EI.

PP-I in Figure 8d on the contrary shows similar behavior to PE-I: For both cycles, the crystallinity increases and peaks at the fourth loop. However, with further processing the degree of crystallinity decreased again and was at its lowest at the last step. The CL-EI shows a steeper incline and a faster decline than the CL-E.

### 3.6. Mechanical Properties of the CL-EI

Due to the availability of multipurpose specimens, only specimens of the processing cycle CL-EI were characterized. As depicted in Appendix A, all four polyolefin grades point to ductile behavior after the first loop with a continuous increase of stress and strain until the specimens yield, followed by a decrease in stress, at least in three of four materials, and by an increasing strain up until failure.

Based on the negligible changes of the PE grades with continuous processing, the results of the second to fifth loop are not presented in this paper. PE-E only shows small changes after the sixth loop due to the CL-EI. Both curves of the PE-I show strain hardening behavior at high strains shortly before failure. After the sixth loop, the materials still exhibit ductile behavior, which leads to the conclusion that the CL-EI has a minimum influence on mechanical properties of the PE grades.

Both PP grades embrittle with continuous processing, which leads to decreasing strain at beak values depicted in Figure 9. This decline is presumably related to chain scission that was also concluded in previous chapters. For both materials, the yield stress after the first to sixth loop stays roughly the same.

## 4. Discussion

Other publications showed that degradation in PE happens by attacking the longer chains and forming crosslinks and additional branches in an uncontrolled manner [14,16]. This formation of longer chains and more interconnected chains leads to a flatter and wider MMD [22], lower values in MFR, an increase in CI due to the formation of carbonyl groups instead of crosslinks or branches [26], lower degrees of crystallinity due to hindered movement of the chains [21,22] and decreasing mechanical properties [19].

In this study, for both PE grades, no significant changes can be seen in Figure 10 for the CL-E with either characterization method (i.e., HPLC, GPC, and MFR). The stabilizers in PE-E showed already in the virgin material very low values. Due to the already low values of the stabilizers and the increased processing steps of the CL-EI, an increase in weight average molar mass started to happen. This increase can be caused by the formation of crosslinks and branches and leads to a decrease of melt flow rate as depicted in the Figure 10 below. Due to longer and connected chains, the movement of the whole melt is restricted, and less melt can flow through the measurement die.

The stabilizers in the PE-I continuously decreased over the course of the loops. However, the existence of stabilizers may be the reason for rather stable weight average molar mass values because the polymer chains are protected for a longer time. Thus, constant melt flow rates were expected and mostly derived.

The results from the IR measurements were not considered in this discussion for either PE type due to falsified values stemming from the migration of lubricants. Even though in this study the results do not add any valuable information, other publications support the assumption that the CI values are a good indicator of degradation [33]. Furthermore, due to negligible effects of the cycles on the mechanical properties of both PEs the results were not added either.

PP has been looked at in multiple publications before this study whereas the typical degradation behavior is chain scission, this usually leads to a higher and slimmer MMD and a decrease in M_w_ [15], therefore to an increase in MFR due to easier movement of the chains and thus unstable melt flow can be obtained. Similar to within the PE grades, the CI should increase by forming carbonyl groups [26]. In some cases, due to degradation the degree of crystallinity can increase which leads to embrittlement and a decrease in mechanical strength [20]. Figure 11 displays the same tendencies for both PP grades in both cycles. The stabilizers decreased with increased processing in both cycles. According to literature [34], the chains of the polymers are protected by the stabilizers; thus, the molar mass should start to be affected after full stabilizer loss. However, as discovered in our work and by Fischer et al. [34], PP starts to degrade immediately. The degradation via chain scission leads to lower weight average molar masses. The polymer chains become shorter and allow for better movement of the melt and with this a lower viscosity which means higher MFR values are to be expected. The PP-I showed the most severe changes in MFR with a value of over 60 g/10 min after the sixth loop. In general, for PP, a good correlation between the results of HPLC, GPC, and MFR was determined.

Similar to the above, distinct values of the CI could not be distinguished for either PP, for this reason no additional diagrams were added. Furthermore, due to non-significant yield stresses and already presented strain values in the chapter above, the mechanical properties are not repeated in this chapter.

## 5. Conclusions

In this study, the degradation behavior of four different polyolefins was investigated. Polymers were chosen based on their chemical composition and on the suggested processing method in their data sheet. Two polyethylenes (PE), a polyethylene high-density (PE-HD) for injection molded products and a polyethylene low-density (PE-LD) for extruded films, and two polypropylenes (PP), one injection molding grade and one extrusion grade, were selected for our research. To investigate the effect of multiple processing loops on the material behavior, the polyolefins were processed with industrially sized machines in two predefined closed-loop cycles. The first cycle comprised of an extrusion and a granulating step, and the second cycle involved additional steps of injection molding and milling. Both cycles were repeated six times. After each of these closed-loops, samples were taken and the respective samples were characterized by high-pressure liquid chromatography, high-temperature gel permeation chromatography, melt flow rate measurements, infrared spectroscopy, differential thermal analysis, and tensile tests.

Investigations with the PE grades showed continuous aging and material degradation was indicated by increasing weight average molar masses and decreasing melt flow rates. Furthermore, the multiple processing loops of the PE types led to a decrease in stabilizer content. The most stable material was the PE-HD injection molding grade. While a significant stabilizer decrease was found in the closed-loop cycle comprising the higher number of processing steps, no significant changes were determined in the other properties especially the CI values, which were distorted, the degrees of crystallinity, with no significant trends, and the mechanical properties, where no relevant changes could be seen. Overall, the results of these investigations point to chain branching and crosslinking within the PE grades.

The degradation behavior of the PP grades was more pronounced than within the PE grades. Both PP materials showed a significant decrease in stabilizer content, a strong decrease in weight average molar masses, an increase in melt flow rate, and embrittlement by a decrease of mechanical strength with increasing cycles. Similar to with the PE grades, the carbonyl index results were falsified, and the degrees of crystallinity did not indicate degradation. Good correlations between the stabilizer contents, weight average molar masses and melt flow rates were found leading to the conclusion that chain scissions dominated for PP in both closed-loop cycles.

## Figures and Tables

**Figure 1 polymers-14-02429-f001:**
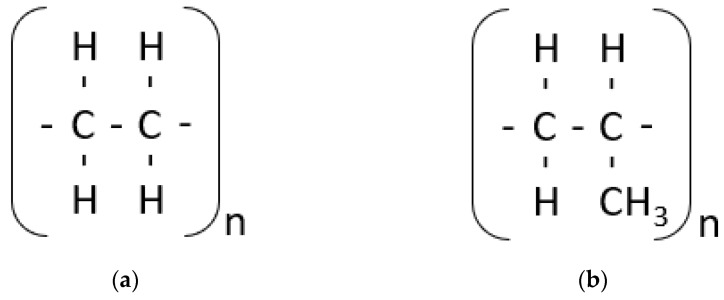
Structural formulas of the analyzed polyolefin grades for (**a**) polyethylene and (**b**) polypropylene.

**Figure 2 polymers-14-02429-f002:**
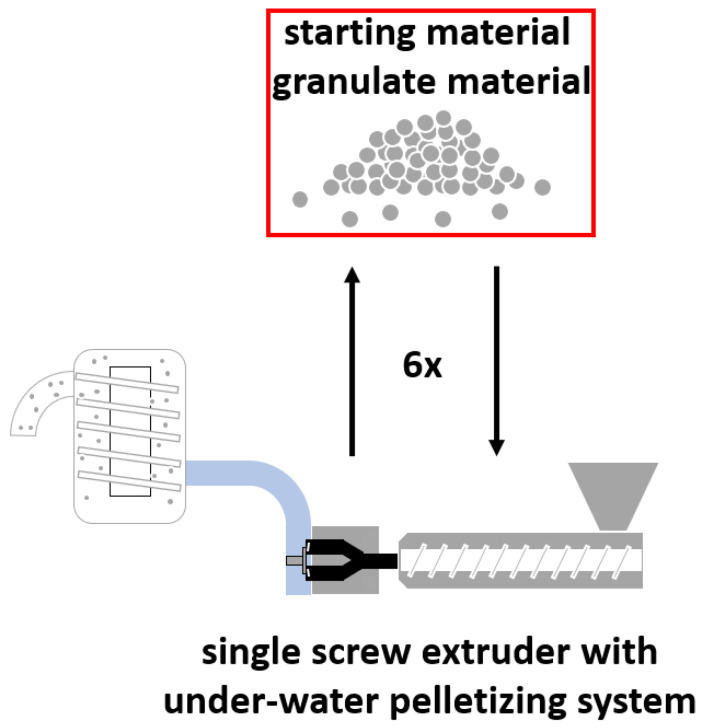
Schematic representation of the extrusion cycle (CL-E) using a 35/34D single screw extruder with an EUP 50 under-water pelletizing system.

**Figure 3 polymers-14-02429-f003:**
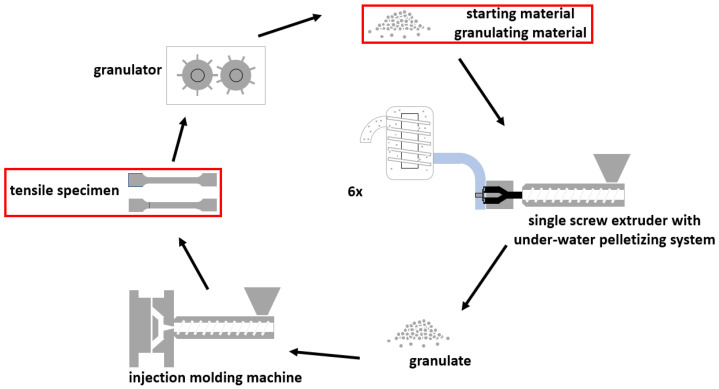
Schematic representation of the extrusion-injection molding cycle (CL-EI) using a 35/34D single screw extruder with an EUP 50 under-water pelletizing system, an injection molding machine victory 60, and granulator S-Max 2 Plus.

**Figure 4 polymers-14-02429-f004:**
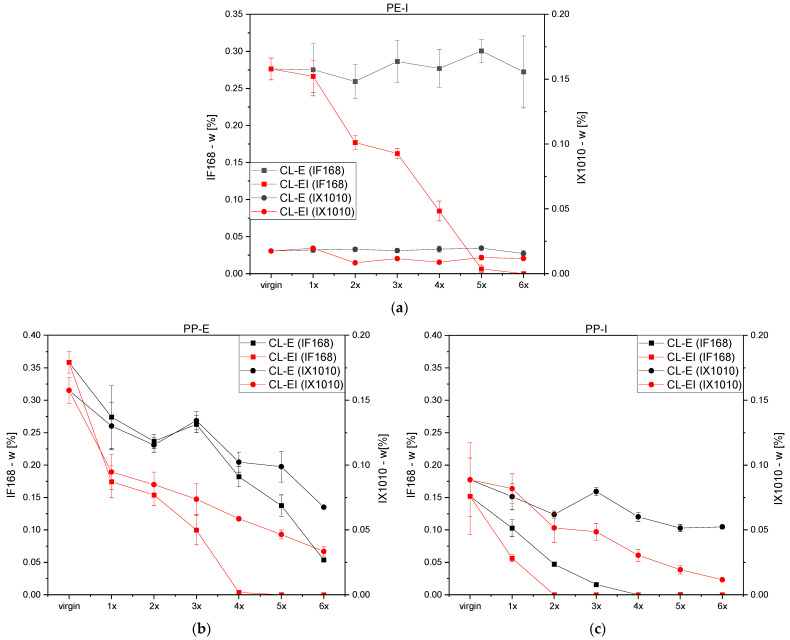
Content of stabilizers IF168 and IX1010 in (**a**) PE-I and both PP grades (**b**) PP-E and (**c**) PP-I after each loop for both recycling cycles.

**Figure 5 polymers-14-02429-f005:**
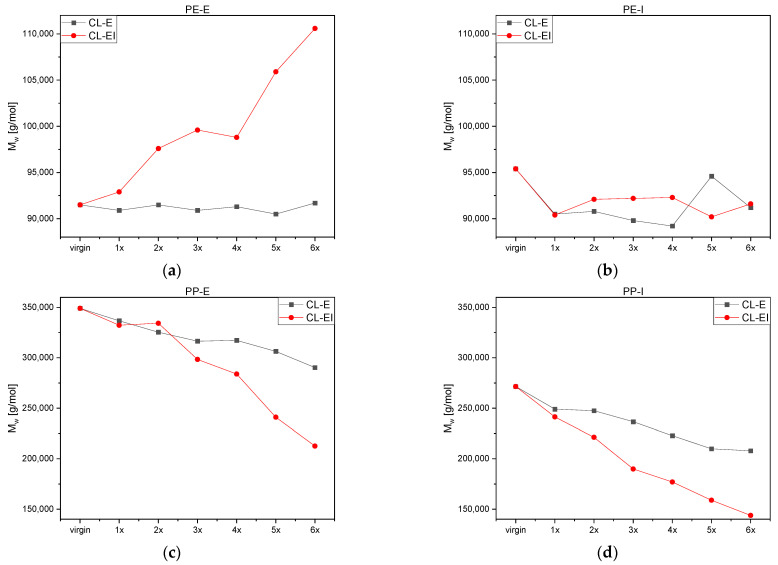
M_w_ after each loop for both processing cycles: (**a**) PE-E, (**b**) PE-I, (**c**) PP-E and (**d**) PP-I.

**Figure 6 polymers-14-02429-f006:**
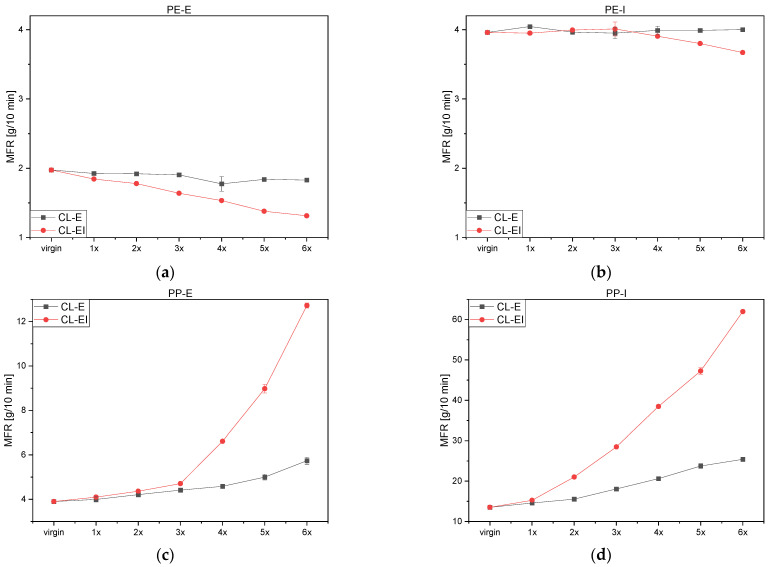
MFR values for both cycles (**a**) PE-E, (**b**) PE-I, (**c**) PP-E and (**d**) PP-I.

**Figure 7 polymers-14-02429-f007:**
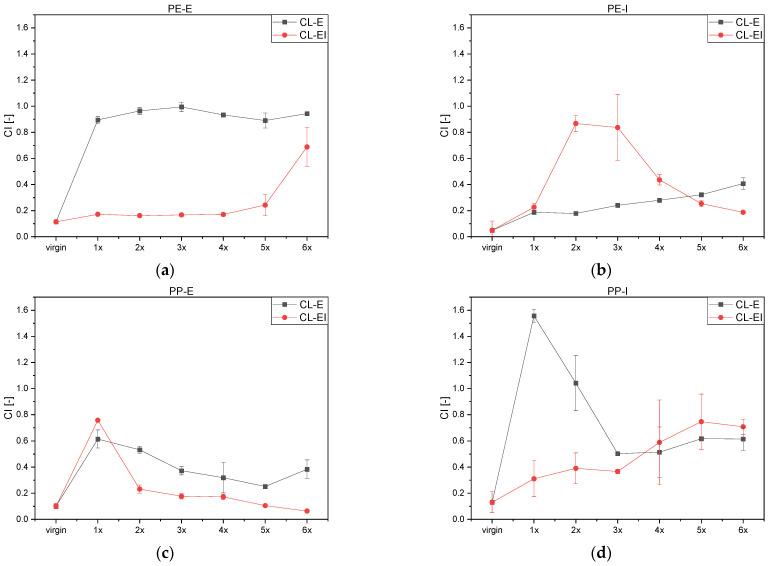
Calculated CI values for (**a**) PE-E, (**b**) PE-I, (**c**) PP-E and (**d**) PP-I.

**Figure 8 polymers-14-02429-f008:**
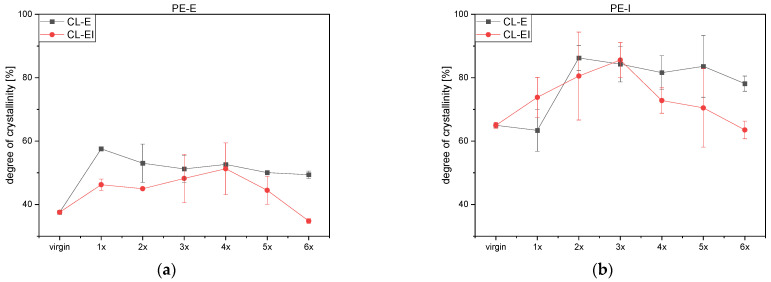
Calculated degrees of crystallinity after each loop for both cycles: (**a**) PE-E, (**b**) PE-I, (**c**) PP-E and (**d**) PP-I.

**Figure 9 polymers-14-02429-f009:**
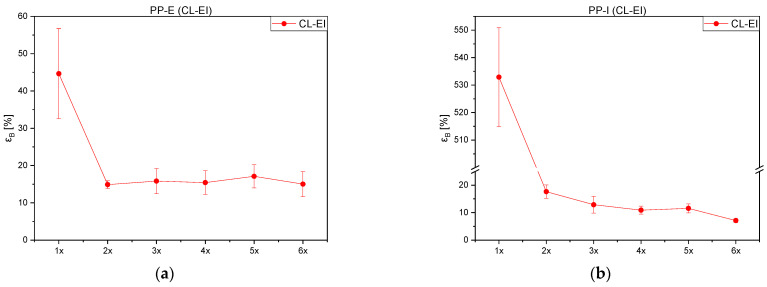
Strain at break values over the course of the processing loops of (**a**) PP-E, and (**b**) PP-I.

**Figure 10 polymers-14-02429-f010:**
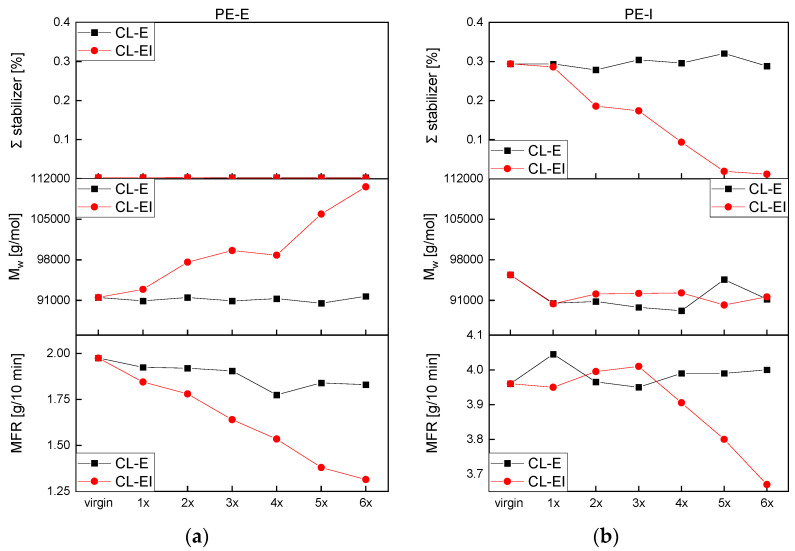
The sum of antioxidants (IF168 and IX1010), the weight average molar mass (M_w_), and the melt flow rate (MFR) as a function of recycling steps for (**a**) PE-E, and (**b**) PE-I.

**Figure 11 polymers-14-02429-f011:**
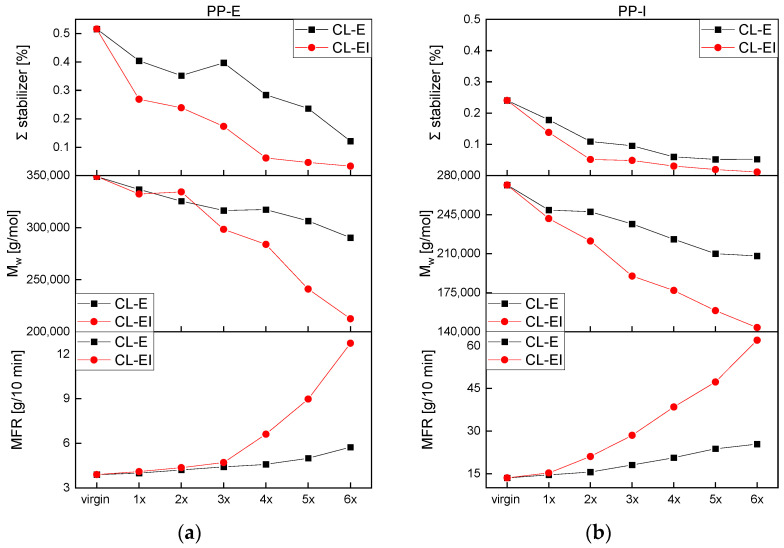
The sum of antioxidants (IF168 and IX1010), the weight average molar mass (M_w_), and the melt flow rate (MFR) as a function of recycling steps for (**a**) PP-E, and (**b**) PP-I.

**Table 1 polymers-14-02429-t001:** Material designation including melt flow rates (MFR), solid densities and application fields.

Characteristic	PE-I	PE-E	PP-I	PP-E
Type	Polyethylene high-density Injection molding	Polyethylene low-density Extrusion	Polypropylene homopolymer Injection molding	Polypropylene homopolymer Extrusion
MFR|g/10 min	4	1.9	12	4
Solid density|kg/m^3^	954	924	905	n.s.
Application	Industrial food transport packaging	Bags, films, flexible packaging, food packaging, liners, pouches	House ware thin wall packaging	Thermoforming packaging

**Table 2 polymers-14-02429-t002:** Temperature program in °C for the heating zones of the single-screw extruder.

Material	FO	Z1	Z2	Z3	AD1	AD2
PE-I	45 °C	190 °C	200 °C	200 °C	200 °C	200 °C
PE-E	45 °C	190 °C	200 °C	200 °C	200 °C	200 °C
PP-I	45 °C	190 °C	220 °C	220 °C	220 °C	220 °C
PP-E	45 °C	190 °C	220 °C	220 °C	220 °C	220 °C

**Table 3 polymers-14-02429-t003:** Temperature program for the injection molding machine.

Material	Die	Z1	Z2	Z3	Z4	Feed
PE-I	220 °C	220 °C	210 °C	205 °C	200 °C	50 °C
PE-E	220 °C	220 °C	210 °C	205 °C	200 °C	50 °C
PP-I	230 °C	230 °C	225 °C	220 °C	210 °C	50 °C
PP-E	230 °C	230 °C	225 °C	220 °C	210 °C	50 °C

**Table 4 polymers-14-02429-t004:** Additional processing parameters of the injection molding machine.

Parameter	Nominal-Values
Clamp force|kN	500
Back pressure|bar	450
Dosage volume|cm^3^	55
Cycle time|s	100
Cooling time|s	20
Back pressure time|s	15

## Data Availability

The data presented in this study are available on request from the corresponding author.

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
