# Peer review of "Determination of the Influence of Multiple Closed Recycling Loops on the Property Profile of Different Polyolefins"

_polymers, 2022, doi:10.3390/polym14122429_

Round 1

Reviewer 1 Report

A manuscript contains original research work, where the authors have presented the results of the study of the influence of multiple recycling loops on the properties of four polyolefin samples including a high-density and low-density polyethylenes and polypropylenes with different melt flow rates. The chemical structure, crystallinity, molecular weight characteristics, melting behavior and mechanical properties of all samples were investigated and analyzed. I recommend accepting current manuscript, however, there are some aspects in this manuscript that should be improved:

1. In the tabs to Figures 11-14, the coordinate axes, especially the abscissa axis, should be labeled. Otherwise, it is not clear by which specific absorption band the carbonyl index was determined.

2. Section 4. Discussion should be supplemented by a discussion of the results of the study of the chemical structure (subpart 3.4) and mechanical strength (subpart 3.6). Since this is not present in the text, the question arises: was it necessary to obtain these results at all?

3. The same applies to section 5. Conclusion.

4. There is an error in the captions to Figures 11(b) and 12(a and b). It should be"...in the wavenumber range from 1750 to 1500 cm-1...".

5. Missing letter in the subpart 3.3.

Reviewer 2 Report

The aim of this paper is a complex study on the influence of multiple recycling cycles on the  properties of polyolefins using  industrial processing machines. The study, which includes five morphological characterization techniques, provides an impressive amount of results. However, there are some parts of the manuscript that should be improved:

1. The specific vibrations highlighted by IR spectra are difficult to identify due to the lack of chemical formula of the analyzed samples.

2. In Figures. 15 (a, b), 17 (a, b) and 18 (a, b), there must be a sign (up, down) indicating the orientation of the endothermic peaks corresponding to the melting temperatures.

3.Chapter 4 (Discussions) must be improved by correlating the obtained results with the chemical and/or structural mechanisms responsible for the evolution of material properties.

4.The conclusions require improvements by correlating all the results obtained.

Round 2

Reviewer 1 Report

The author took into account all the comments of the reviewer. The text has been corrected and supplemented in accordance with the reviewer' comments. The article can be accepted for publication.

Author Response

Dear Reviewer, thank you for your recommendation to publish.